# Pre-training with Random Orthogonal Projection Image Modeling

**Maryam Haghighat**[*,†,††]**, Peyman Moghadam**[§,†]**, Shaheer Mohamed**[§,†]**, Piotr Koniusz**[*,§,‡]

[§]Data61♥CSIRO   [†]Queensland University of Technology   [‡]Australian National University
[†]`name.lastname@qut.edu.au`, [§]`name.lastname@data61.csiro.au`

## Abstract

Masked Image Modeling (MIM) is a powerful self-supervised strategy for visual pre-training without the use of labels. MIM applies random crops to input images, processes them with an encoder, and then recovers the masked inputs with a decoder, which encourages the network to capture and learn structural information about objects and scenes. The intermediate feature representations obtained from MIM are suitable for fine-tuning on downstream tasks. In this paper, we propose an Image Modeling framework based on random orthogonal projection instead of binary masking as in MIM. Our proposed Random Orthogonal Projection Image Modeling (ROPIM) reduces spatially-wise token information under guaranteed bound on the noise variance and can be considered as masking entire spatial image area under locally varying masking degrees. Since ROPIM uses a random subspace for the projection that realizes the masking step, the readily available complement of the subspace can be used during unmasking to promote recovery of removed information. In this paper, we show that using random orthogonal projection leads to superior performance compared to crop-based masking. We demonstrate state-of-the-art results on several popular benchmarks.

## 1 Introduction

Masked Image Modeling (MIM) (Bao et al., 2022; He et al., 2022; Xie et al., 2022) has achieved promising performance by pre-training backbones that are then fine-tuned on different downstream tasks such as image classification or semantic segmentation.

Most MIM techniques follow the general paradigm of self-prediction, *i.e.*, they randomly mask out some regions in the input data and then learn to recover the missing data. Current MIM methods (Bao et al., 2022; He et al., 2022; Xie et al., 2022) mainly apply masking in the spatial domain by randomly excluding image patches. Since raw image pixels are highly correlated within their spatial neighbourhood, a high masking ratio (60%-75%) leads to high quality features (He et al., 2022; Xie et al., 2022).

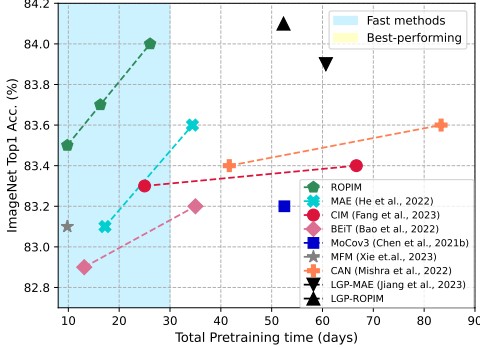

Figure 1: Training efficiency of ROPIM *vs.* other methods. ROPIM achieves a higher accuracy (see also LGP-ROPIM) with a lower training time. The blue and yellow regions indicate fast methods and high-accuracy methods, respectively. ROPIM has both high accuracy and is fast (the green region).

Existing MIM approaches typically replace a random set of input tokens with a special learnable symbol, called MASK, and aim to recover either masked image pixels (He et al., 2022; Xie et al., 2022), masked content features (Wei et al., 2022) or latent representations (Baevski et al., 2022). This additional learnable MASK token is applied over large masked areas, up to 75% of image (He et al., 2022), and is not used in the fine-tuning stage (Fang et al., 2023).

In this paper, we propose a new Random Orthogonal Projection Image Modeling (ROPIM) pre-training framework, which uses a simple projection strategy with provable noise bounds due to the loss of information. ROPIM is based on orthogonal projection (Charikar et al., 2002) which is applicable to raw pixels and latent feature representations. Figure 1 compares the top-1 accuracy

---

[*]Corresponding authors.   [††]MH conducted this work during the employment with CSIRO.   PK also in charge of the theory.   The code is available at `https://github.com/csiro-robotics/ROPIM`.

Figure 2: Our proposed Random Orthogonal Projection Image Modeling (ROPIM) *vs*. Masked Image Modeling (MIM). MIM in Fig. 2a performs masking on patches of an input image, passed to the backbone, followed by unmasking. Our ROPIM in Fig. 2b performs the orthogonal projection of patch embeddings onto a random subspace, passed to the backbone, followed by application of the complement of orthogonal projection. Thus, the loss focuses on the recovery of the lost information.

*vs*. total pre-training (PT) time with sate-of-the-art SSL methods. ROPIM achieves higher accuracy while requiring significantly less total pre-training time. Total PT time is calculated as time per epoch multiplied by number of epochs. For fair comparisons, the reported times in Figure 1 are derived from the use of the same resources (8×P100 GPUs) and maximum possible batch size per GPU for each method. More details are discussed in Table 5 of Appendix A.

Figure 2 shows our ROPIM approach. Our framework does not require a separate tokenizer network as in BEiT (Bao et al., 2022) and CIM (Fang et al., 2023) or a large decoder that requires additional computations. Figure 3 shows that no matter whether an input image is randomly masked or projected to an orthogonal subspace, the network is encouraged to recover its complement. Adding masked/projected image to its complement

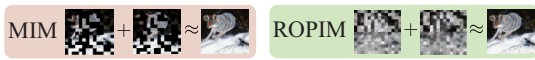

Figure 3: For MIM, unmasked parts of the recovered image, combined with the masked parts do approximate the input image. Our tokens, randomly projected and complement of the projection (equivalent of unmasking) along spatial modes, also approximately recover the input when added together.

subspace has to approximate the original image. ROPIM projects the features of patch embeddings along their spatial mode into a random subspace. Subsequently, we use the complement of this random subspace to guide the loss function to recover the removed information. We apply Random Orthogonal Projection (ROP) at the token level, hence the imposed computation overhead is negligible. We note that our proposed approach does not require MASK tokens.

Figure 4 compares visually binary masking in MIM with Random Orthogonal Projection (ROP). Compared with ROP, binary masking creates limited number of patterns, *e.g.*, for 4 tokens one gets $2^4$ masking and unmasking patterns only. Such a randomness is limited–the network cannot learn to recover from masking patterns that never occurred. In contrast, ROP is a linear interpolation between several tokens. Thus, it can be considered as a

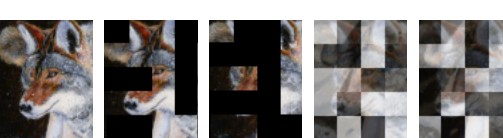

Figure 4: Left to right: original image, masking, unmasking, ROP, complement of ROP. Notice the "continuous" masking nature of *ROP* and *complement of ROP*.

"continuous" masking where multiple locations are combined into a coefficient by the projection pattern. Since this "combination" is achieved by the projection matrix, we readily have the complement space needed for recovery of the removed information via a lightweight projection step. Hence, the network learns faster (Fig. 1) as it is challenged by richer masking-unmasking patterns. Moreover, ROP is a form of randomized data corruption, or rather a lossy projection step with a guaranteed bound on the noise variance it introduces. In contrast, binary masking in MIM methods is prone to remove crucial image regions, potentially resulting in performance degradation (Li et al., 2021), especially when high masking ratio is applied. Injecting a bounded noise is hence critical to learn semantically meaningful features.

Our contributions can be summarized as follows:

i. We propose ROPIM, a simple but effective image modeling based on the so-called count sketching, with the aim of reducing local semantic information under the bounded noise variance.

ii. In contrast to the binary masking (MIM), ROP forms "continuous" masking by a known projection matrix which has an easily-obtainable complement space matrix, which we use in the reconstruction loss to guide the recovery of the removed input information.

iii. We propose to project patch tokens along their spatial mode into a random subspace, which is computationally negligible, enjoying the high throughput of MIM methods.

Our results show that proposed "continuous" masking/unmasking strategy creates a rich model with less pre-training cost, without the use of an auxiliary network, large decoder or a tokenizer.

## 2 RELATED WORK

**Transformers** (Vaswani et al., 2017), popular in natural language processing (BERT (Devlin et al., 2018) and GPT-3 (Brown et al., 2020)), capture attention between tokens. Image Transformers (Parmar et al., 2018) and Vision Transformers (ViT) (Dosovitskiy et al., 2021) also achieve persuasive results in supervised and unsupervised learning on large-scale datasets. ViT inspired data-efficient models such as DeiT (Touvron et al., 2021), self-supervised DINO (Caron et al., 2021b), CrossViT (Chen et al., 2021a), and general-purpose architectures such as Swin-T (Liu et al., 2021b) and Twins (Chu et al., 2021). Kindly notice, in this work, we do not propose new transformers.

**Self-supervised Learning** (Liu et al., 2021a) is essential for data hungry architectures with transformers. The lack of labels has led the vision community to study self-supervised learning based on contrastive or generative setting. Many self-supervised models use pretext tasks (Liu et al., 2021a). Generative techniques such as Denoising AutoEncoder (DAE) (Vincent et al., 2008) inject noise into the input data and train a network with a bottleneck to recover the original input. Many methods build on DAE under different corruption strategies, *e.g.*, masking pixels or removing color channels. Contrastive models such as DINO and MoCo v3 (Caron et al., 2021a; Chen et al., 2021b) use data augmentations to generate different image views, pull positive feature pairs while pushing away negative pairs. BYOL (Grill et al., 2020) and SimSiam (Chen & He, 2021b) eliminate negative sampling and prevent dimensional collapse. COSTA (Zhang et al., 2022) eliminates multiple views. Finally, COLES (Zhu et al., 2021), EASE (Zhu & Koniusz, 2022a) and GLEN (Zhu & Koniusz, 2022b) introduce the negative sampling into Laplacian Eigenmaps.

**Masked Image Modeling (MIM)** techniques (Bao et al., 2022; He et al., 2022; Xie et al., 2022) learn representations from images corrupted by masking. Inspired by success in transformer-based masked language modeling (Devlin et al., 2018), Dosovitskiy *et al.* (Dosovitskiy et al., 2021) explored the prediction of masked image patches for self-supervision for visual data. Recent works (Bao et al., 2022; He et al., 2022; Xie et al., 2022; Baevski et al., 2022; Wei et al., 2022) use MIM with a transformer-based architecture (Vaswani et al., 2017) and various objective functions.

Most MIM methods (Bao et al., 2022; He et al., 2022; Xie et al., 2022; Baevski et al., 2022; Wei et al., 2022; Mishra et al., 2022) use masking in the spatial domain by randomly excluding image patches or tokens. MAE (He et al., 2022) and SimMIM (Xie et al., 2022) recover masked raw pixels. BEiT (Bao et al., 2022) uses a discrete VAE (dVAE) network to transform image patches to visual tokens. During pre-training, the semantic tokens are recovered. However, BEiT requires an additional dVAE network to be pre-trained on patches. iBOT (Zhou et al., 2022) uses a teacher network online tokenizer and performs self-distillation on masked patch and class tokens. Data2vec (Baevski et al., 2022) uses a teacher-student framework to reconstruct latent representations. MaskedFeat (Wei et al., 2022) recovers Histograms of Oriented Gradients. Tian *et al.* (Tian et al., 2022) use different learning objectives for image degradation, including zoom-in, zoom-out, fish-eye distortion, blur and de-colorization. MFM (Xie et al., 2023) uses Fast Fourier Transform for masked frequency modeling. Recent approaches CAN (Mishra et al., 2022) and LGP (Jiang et al., 2023) combine Contrastive Learning (CL) with MIM for due to their complementarity. LGP (Jiang et al., 2023) implements layer grafted pre-training in a sequential fashion. Corrupted Image Modeling (CIM) (Fang et al., 2023) uses an auxiliary generator with a trainable BEiT network to corrupt images in a better way than artificial MASK tokens. Similar to BEiT, CIM requires an additional dVAE network–its pre-training per epoch is $2\times$ slower than BEiT (Fang et al., 2023).

Our ROPIM differs from the above models: (i) we do not mask patches, but perform projection onto a random subspace along the spatial mode of tokens, (ii) we do not perform unmasking, but use the complement of the subspace to support the loss of ROPIM in recovering the removed information.

**Count Sketching** is a widely used unsupervised dimensionality reduction technique (Weinberger et al., 2009; Cormode & Muthukrishnan, 2005). Several variants of count sketching have been proposed in the literature, including Count-Min Sketch (Cormode & Muthukrishnan, 2005). However, the core concept is based on capturing a small sketch of the data with a random projection function.

# 3 APPROACH

Below, we detail our ROPIM pipeline. Firstly, we explain our notations and ROP. Then, we introduce our problem formulation and ROPIM pipeline.

## 3.1 PRELIMINARIES

**Notations.** Let $\mathbf{x} \in \mathbb{R}^d$ be a $d$-dimensional feature vector. $\mathcal{I}_N$ stands for the index set $\{1, 2, \cdots, N\}$. We define $\mathbf{1} = [1, ..., 1]^T$ ('all ones' vector). Capitalised bold symbols such as $\mathbf{\Phi}$ denote matrices, lowercase bold symbols such as $\phi$ denote vectors, and regular fonts denote scalars *e.g.*, $\Phi_{i,j}$, $\phi_i$, $n$ or $Z$. $\Phi_{i,j}$ is the $(i, j)$-th entry of $\mathbf{\Phi}$. Symbol $\delta(x) = 1$ if $x = 0$, $\delta(x) = 0$ if $x \neq 0$, and $\mathbb{I}$ is the identity matrix. Operator $\|\cdot\|_1$ on matrix is the $\ell_1$ norm of vectorised matrix.

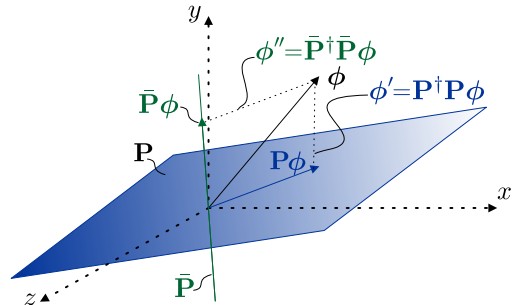

Figure 5: Understanding the projection of $\phi$ on the unitary projection matrix $\mathbf{P}$ (subspace), given as $\mathbf{P}\phi$, and its retraction given as $\phi' = \mathbf{P}^\dagger\mathbf{P}\phi$. Projection matrix $\bar{\mathbf{P}}$ (subspace) complementary to $\mathbf{P}$ is also indicated. Vector $\phi$ projected on $\bar{\mathbf{P}}$ and then retracted from it is given as $\phi'' = \bar{\mathbf{P}}^\dagger\bar{\mathbf{P}}\phi$. Notice that $\phi' + \phi'' = \phi$. The lossy nature of this projection occurs when $\mathbf{P}^\dagger\mathbf{P} + \bar{\mathbf{P}}^\dagger\bar{\mathbf{P}} \neq \mathbb{I}$, *i.e.*, not the full diagonal matrix is recovered.

**Proposition 1.** *Let $K$ and $K'$ be the sizes of the input and the projected output. Let vector $\mathbf{h} \in \mathcal{I}_{K'}^K$ contain $K$ uniformly drawn integer numbers from $\{1, \cdots, K'\}$ and vector $\mathbf{s} \in \{-1, 1\}^K$ contain $K$ uniformly drawn values from $\{-1, 1\}$. The projection matrix $\mathbf{P} \in \{-1, 0, 1\}^{K' \times K}$ is given as $P_{ij}(\mathbf{h}, \mathbf{s}) = s_j \cdot \delta(h_j - i)$ and the projection $\Pi : \mathbb{R}^K \to \mathbb{R}^{K'}$ is a linear operation $\Pi_{\mathbf{h}, \mathbf{s}}(\phi) = \mathbf{P}(\mathbf{h}, \mathbf{s})\phi$ (or simply $\Pi(\phi) = \mathbf{P}\phi$).*

Following are properties of count sketches we utilize in our work:

**Property 1.** *The inner product of count sketches is an unbiased estimator. Specifically, we have $\mathbb{E}_{\mathbf{h}, \mathbf{s}}\left[\langle\Pi_{\mathbf{h}, \mathbf{s}}(\phi_x), \Pi_{\mathbf{h}, \mathbf{s}}(\phi_y)\rangle - \langle\phi_x, \phi_y\rangle\right] = 0$ with variance bounded by $\frac{1}{K'}(\langle\phi_x, \phi_y\rangle^2 + \|\phi_x\|_2^2\|\phi_y\|_2^2)$.*

*Proof.* See Weinberger *et al.* (Weinberger et al., 2009) for proof. □

**Property 2.** *The unitary projection matrix $\mathbf{P}$ enjoys a simple pseudo-inverse $\mathbf{P}^\dagger = \frac{K'}{K}\mathbf{P}^T$.*

*Proof.* The transpose for inverse follows from the fact that $\mathbf{P}$ is constructed as a unitary matrix. □

**Property 3.** *The distance of vector $\phi$ to subspace $\mathbf{P}$ is given as $\|\phi - \mathbf{P}^\dagger\mathbf{P}\phi\|_2$. Thus $\phi' = \mathbf{P}^\dagger\mathbf{P}\phi$ is the vector with the removed information resulting from the lossy operations: (i) projection of $\phi$ on subspace $\mathbf{P}$ followed by (ii) retraction from the subspace into the original feature space.*

*Proof.* These relations follow from Grassmann feature maps of subspaces (Harandi et al., 2015). □

**Property 4.** *As the complement of $\mathbf{P}^\dagger\mathbf{P}$ is $\mathbb{I} - \mathbf{P}^\dagger\mathbf{P}$, the distance of $\phi$ to the complement basis of subspace $\mathbf{P}$ is $\|\mathbf{P}^\dagger\mathbf{P}\phi\|_2$. Thus $\phi'' = (\mathbb{I} - \mathbf{P}^\dagger\mathbf{P})\phi$ is a vector complementary to $\phi'$, i.e., $\phi' + \phi'' = \phi$.*

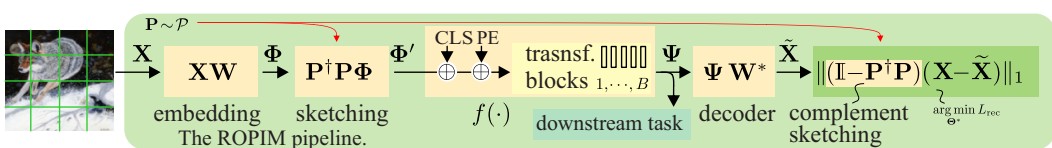

Figure 6: Overview of the Random Orthogonal Projection Image Modeling (ROPIM) pipeline. An image is divided into patch tokens, and embedded. Sketching matrix $\mathbf{P} \sim \mathcal{P}$ is drawn and ROP (with its inverse) is applied to embeddings $\mathbf{\Phi}$. Subsequently, $\mathbf{\Phi}'$ is passed through transformer $f(\cdot)$. Operator $\oplus$ is an addition. CLS and PE are the class token and positional embedding. Finally, decoder (only one linear projection layer) and the reconstruction loss which targets the inverse projection, are applied. Once ROPIM is trained, we use $\mathbf{\Psi}$ as feature representations for the downstream task.

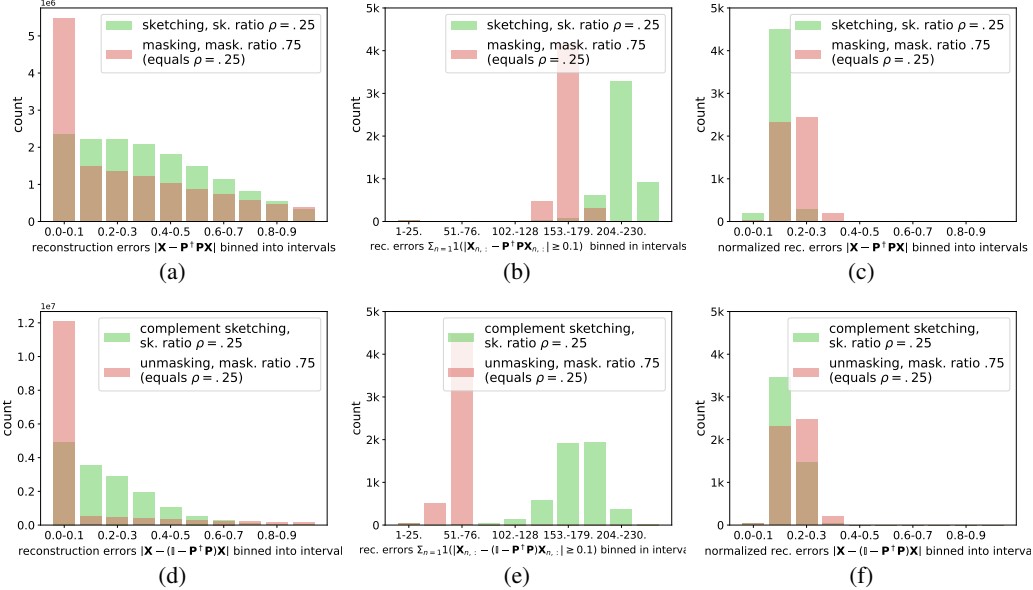

Figure 7: (*top*) Comparison of errors of ROP by sketching *vs*. masking. 5000 images (normalized in range [0, 1]) randomly sampled from CIFAR10 were divided each into $16 \times 16$ tokens. Sketching ratio $\rho = .25$ was applied which corresponds to masking ratio $1-\rho = .75$. Fig. 7a shows histogram-binned $\ell_1$ errors (green) computed between tokens in $\mathbf{X}$ and their sketch projected-retracted lossy versions $\mathbf{P}^\dagger \mathbf{P} \mathbf{X}$. The $\ell_1$ errors (red) between tokens in $\mathbf{X}$ and the masked versions are shown. As is clear, masking produces many locations with zero error. In contrast, sketching introduces some errors to every token. Fig. 7b shows histogram counts of tokens for which the reconstruction error was greater than 0.1. Clearly, sketching modified more regions than masking. Fig. 7c is as 7a but the reconstruction error of each token is normalized by the number of tokens in image with error greater than 0.1. Clearly, sketching introduces lesser error per token but modifies plenty more spatial locations than masking, which explains why ROP is superior to masking. (*bottom*) The same analysis for complement sketching and unmasking. Fig. 7d, 7e and 7f again show that ROP operates on more regions spatially-wise than unmasking. Unlike other operations, ROP enjoys an easy complement sketching in analogy to unmasking.

> Figure 5 illustrates the essence of Properties 3 & 4. Projection onto $\mathbf{P}$ is a lossy operation whose lost information is quantified by the bounded noise variance in Property 1. The same applies to projection onto $\bar{\mathbf{P}}$, *i.e.*, the complement of $\mathbf{P}$. Figure 7 shows errors computed on $16 \times 16 \times 5000$ image tokens (CIFAR10). Unlike masking/unmasking, ROPIM "soft-masks" & "soft-unmasks" more tokens at once by principled operations with bounded reconstruction errors.

## 3.2 PROBLEM FORMULATION

We employ the standard vision transformer (Dosovitskiy et al., 2021) for our Random Orthogonal Projection Image Modelling (ROPIM). Figure 6 provides an overview of the ROPIM pipeline.

For an image with dimensions $H \times W \times C$ representing height, width, and the number of channels, we extract a series of 2D patches, reshape them into vectors, and stack into a matrix $\mathbf{X} \in \mathbb{R}^{N \times (P^2 \cdot C)}$, where $P \times P$ is the patch size, and $N$ is the number of patches extracted with the goal of forming patch embeddings. Patch embeddings are obtained as:

$$\mathbf{\Phi} = \mathbf{X}\mathbf{W}, \tag{1}$$

where $\mathbf{W} \in \mathbb{R}^{(P^2 \cdot C) \times D}$ is the linear projection matrix used to obtain the matrix of em-

---

**Algorithm 1** Random Orthogonal Projection Image Modeling (ROPIM).

---

**Input** $\mathcal{D}_{\text{train}}$: training dataset; $\tau$: iterations; $\rho$: sketching ratio; set $K' = \rho K$.
**for** $t = 1, \cdots, \tau$ **do**
     $\mathbf{X} \sim \mathcal{D}_{\text{train}}$ (draw an image with tokens)
     $\mathbf{P} \sim \mathcal{P}(K')$ (draw proj. matrix (Propos. 1))
     Update the main network branch by Eq. (4):
         $\underset{\mathbf{\Theta}^*}{\arg\min} \, L_{\text{rec}}(\mathbf{X}; \mathbf{\Theta}^*, \mathbf{P})$
**end for**

---

beddings $\mathbf{\Phi} \in \mathbb{R}^{N \times D}$. In MIM, a random portion of the input patch embeddings are replaced with a MASK token, and then a network is trained to recover the masked patches.

In this paper, instead of patch-wise masking, we apply ROP with the sketching ratio $\rho = \frac{K'}{K}$ which determines the lossy effect on projected embeddings. We apply the ROP operation along the spatial mode of the matrix of embeddings. Specifically, we perform the projection matrix followed by retraction, as explained in Property 3:
$$\mathbf{\Phi}' = \mathbf{P}^{\dagger}\mathbf{P}\mathbf{X}\mathbf{W}, \tag{2}$$

where $\mathbf{P}$ is the unitary projection matrix with a trivial pseudo-inverse (Property 2). Matrix $\mathbf{\Phi}'$ contains our projected embeddings of patches. Note $\mathbf{\Phi}' \in \mathbb{R}^{N \times D}$ & $\mathbf{\Phi} \in \mathbb{R}^{N \times D}$ have the same size.

Then we add the class token and positional embeddings $\mathbf{\Phi}'$, and we pass $\mathbf{\Phi}'$ into the transformer, $i.e.$, $\mathbf{\Psi} = f(\mathbf{\Phi}')$. We use a linear prediction head to reconstruct the raw pixel values via the $\ell_1$ loss. Specifically, we apply:
$$\widetilde{\widetilde{\mathbf{X}}} = (\mathbb{I} - \mathbf{P}^{\dagger}\mathbf{P})f(\mathbf{\Phi}')\mathbf{W}^*, \tag{3}$$

where $\mathbf{W}^* \in \mathbb{R}^{D \times (P^2 \cdot C)}$ is the output linear projection, and $(\mathbb{I} - \mathbf{P}^{\dagger}\mathbf{P})\mathbf{\Psi}$ is the complement map of $\mathbf{P}^{\dagger}\mathbf{P}$ (Property 4) which explicitly promotes the recovery of the removed input information. Figure 8 illustrates the effects of Eq. (2) & (3) on images and patches within. We skip $\mathbf{W}$ & $\mathbf{W}^*$ for clarity.

Combining the above steps from Eq. (2) and (3), we obtain the ROPIM pipeline:

$$L_{\text{rec}}(\mathbf{X}; \mathbf{\Theta}^*, \mathbf{P}) = \|(\mathbb{I} - \mathbf{P}^{\dagger}\mathbf{P})(\mathbf{X} - \widetilde{\mathbf{X}})\|_1, \tag{4}$$
$$\widetilde{\mathbf{X}} = f_{\mathbf{\Theta}}(\mathbf{P}^{\dagger}\mathbf{P}\mathbf{X}\mathbf{W})\mathbf{W}^*. \tag{5}$$

Notice that $\widetilde{\widetilde{\mathbf{X}}} = (\mathbb{I} - \mathbf{P}^{\dagger}\mathbf{P})\widetilde{\mathbf{X}}$ but we move the complement ROP (with its inverse), $\mathbb{I} - \mathbf{P}^{\dagger}\mathbf{P}$, into Eq. (4) as the complement ROP (with its inverse) has to be also applied to $\mathbf{X}$ in order to promote only the recovery of lost information. $L_{\text{rec}}(\mathbf{X}; \mathbf{\Theta}^*, \mathbf{P})$ is the reconstruction loss we minimize w.r.t. $\mathbf{\Theta}^* \equiv \{\mathbf{\Theta}, \mathbf{W}, \mathbf{W}^*\}$, that is, network parameters $\mathbf{\Theta}$, and linear projection matrices $\mathbf{W}$ and $\mathbf{W}^*$. Notice that $\widetilde{\mathbf{X}}$ depends on several arguments, $i.e.$, $\widetilde{\mathbf{X}} \equiv \widetilde{\mathbf{X}}(\mathbf{X}; \mathbf{\Theta}^*, \mathbf{P})$ but we drop them in subsequent equations for brevity.

ROPIM is given by Alg. 1. We skip mini-batch level operations for simplicity. Note that for each image sample, we draw a new projection matrix $\mathbf{P}$ according to Proposition 1 and then we simply minimize the reconstruction loss from Eq. (4).

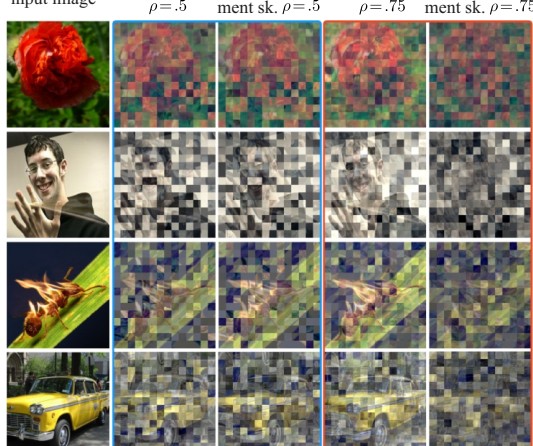

Figure 8: The effect of ROP on images sampled from ImageNet (first column). Second/fourth columns: images after applying Eq. (2) with $\mathbf{W} = \mathbb{I}$, $i.e.$, $\mathbf{P}^{\dagger}\mathbf{P}\mathbf{X}$ (sketching ratio $\rho = .5$ and $\rho = .75$ respectively). Third/fifth columns: images after applying the complement of Eq. (2), $i.e.$, $(\mathbb{I} - \mathbf{P}^{\dagger}\mathbf{P})\mathbf{X}$ (for $\rho = .5$ and $\rho = .75$ respectively). Notice that adding images in columns two and three (or four and five) recover the original images.

## 4 EXPERIMENTS

### 4.1 DATASETS

We perform self-supervised pre-training on ImageNet-1k (Russakovsky et al., 2015). For further ablation studies we use ImageNet100 (Tian et al., 2020) to pre-train a smaller variant of ViT. We also use iNaturlaist 2017 (Van Horn et al., 2018) classification dataset and ADE20k segmentation dataset (Zhou et al., 2019) for large-scale networks and datasets evaluations. Flowers102 (Nilsback & Zisserman, 2008), CUB-200 (Wah et al., 2011) and CIFAR10/100 are used for evaluation of smaller scale experiments discussed in Appendix B, Table 6.

**ImageNet-1K** (Russakovsky et al., 2015) used by us is ILSVRC-2012 with 1k classes and 1.3M images. **ADE20K** (Zhou et al., 2019) is a semantic segmentation dataset including 150 semantic categories, 20K training images, 2K validation images, and 3K images for testing. **iNaturalist 2017**

**(iNat17)** dataset (Van Horn et al., 2018) contains images from 5089 fine-grained categories of different species of plants and animals. Those categories are annotated with 13 super-categories including 579,184 training images and 95,986 validation images. **CIFAR10/CIFAR100** (Krizhevsky et al., 2009) consists of 50K and 10K training and testing images of resolution 32×32 from 10 and 100 classes respectively. **ImageNet100** is a subset of ImageNet Large Scale Visual Recognition Challenge 2012. It contains random 100 classes proposed by Tian *et al.* (Tian et al., 2020). ImageNet100 train and validation sets contain 1300 and 50 images per class, respectively.

## 4.2 EXPERIMENTAL SETTINGS

We conduct our experiments on ImageNet-1k with ViT-Base (ViT-B) and ViT-Small (ViT-S) (Dosovitskiy et al., 2021). For further ablation studies we use ViT-Tiny (ViT-T) (Dosovitskiy et al., 2021) which includes 12 layers, 3 heads and embedding size 192 and a total 5.6M number of parameters. The patch size of all ViT models is 16×16 indicated by '/16' In all of our experiments we use relative positional embedding (Dosovitskiy et al., 2021).

We train our models using AdamW optimizer, a weight decay of 0.05, $\beta_1 = 0.9$, $\beta_2 = 0.95$, and a cosine learning rate scheduler. ViT-B and ViT-S are pre-trained with an initial 10 epochs linear warm-up procedure and a batch size of 1520. For ROPIM, sketching ratio $\rho = \frac{1}{7}$ is used unless otherwise mentioned.

Table 1: Top-1 classification accuracy on ImageNet-1k. *BEiT and CIM need an additional stage to pre-train dVAE tokenizer.

| Method | Backbone | Pre-training epochs | Fine-tuning Top1 acc % |
|---|---|---|---|
| Supervised (Touvron et al., 2021) | ViT-B/16 | | 81.8 |
| DINO (Caron et al., 2021a) | ViT-B/16 | 1600 | 82.8 |
| MoCo v3 (Chen et al., 2021b) | ViT-B/16 | 600 | 83.2 |
| BEiT* (Bao et al., 2022) | ViT-B/16 | 300 (+dVAE) | 82.9 |
| BEiT* (Bao et al., 2022) | ViT-B/16 | 800 (+dVAE) | 83.2 |
| MAE (He et al., 2022) | ViT-B/16 | 800 | 83.1 |
| MFM (Xie et al., 2023) | ViT-B/16 | 300 | 83.1 |
| CIM-RESPIX* (Fang et al., 2023) | ViT-B/16 | 300 (+dVAE) | 83.3 |
| CIM-REVDET* (Fang et al., 2023) | ViT-B/16 | 300 (+dVAE) | 83.3 |
| ROPIM | ViT-B/16 | 300 | **83.5** |
| ROPIM | ViT-B/16 | 500 | **83.7** |
| ROPIM | ViT-B/16 | 800 | **84.0** |
| Supervised (Touvron et al., 2021) | ViT-S/16 | | 79.9 |
| DINO (Caron et al., 2021a) | ViT-S/16 | 1600 | 81.5 |
| MoCo v3 (Chen et al., 2021b) | ViT-S/16 | 600 | 81.4 |
| BEiT* (Bao et al., 2022) | ViT-S/16 | 300 (+dVAE) | 81.3 |
| CIM-RESPIX* (Fang et al., 2023) | ViT-S/16 | 300 (+dVAE) | 81.5 |
| CIM-REVDET* (Fang et al., 2023) | ViT-S/16 | 300 (+dVAE) | 81.6 |
| ROPIM | ViT-S/16 | 300 | **81.8** |
| ROPIM | ViT-S/16 | 500 | **82.0** |

After pre-training, we evaluate our models on image classification and segmentation benchmarks with end-to-end fine-tuning. Detailed hyperparameters are available in Appendix D. DAE representations, *e.g.* MIM, are strong nonlinear features and perform well when a nonlinear head is tuned (He et al., 2022). On the other hand, linear probing results of contrastive SSL methods are not well correlated with their transfer learning performance (Chen & He, 2021a). Therefore, similar to prior work (Fang et al., 2023; He et al., 2022; Xie et al., 2022) fine-tuning results are the main focus of this paper.

## BASELINES

We compare our ROPIM against several state-of-the-art self-supervised pre-training methods including DINO (Caron et al., 2021a), MoCo v3 (Chen et al., 2021b), BEiT (Bao et al., 2022), MAE (He et al., 2022), MFM (Xie et al., 2023) and CIM (Fang et al., 2023). We also included models trained in a supervised setting, denoted as "Supervised", where a classification or segmentation head is used for training. Notice that the "Supervised" baselines do not use the image decoder.

## 4.3 COMPARISON TO THE STATE OF THE ART

Table 1 shows the comparison of our ROPIM method with the current self-supervised pre-training approaches on ImageNet-1k. For a fair comparison, the results are reported for a similar backbone, *i.e.* ViT-B or ViT-S. Using ViT-B, our approach achieves a top-1 classification accuracy of 83.5% and 83.7% for 300 and 500 pre-training epochs respectively, outperforming all other baselines without requiring an additional dVAE training to be used as the tokenizer network or a large decoder.

### 4.3.1 TRANSFER LEARNING

To evaluate the pre-trained models for transfer learning[1], we study the performance of our pre-trained ViT-B model on two large-scale classification and segmentation datasets, *i.e.* iNaturlaist 2017 and ADE20K.

**Classification**. Table 2 shows the classification accuracy of iNaturalist 2017, CIFAR10 and CIFAR100 when fine-tuning the model pre-trained on ImageNet-1k. Compared to the reported accuracy in MAE (He et al., 2022) with the same backbone, i.e. ViT-B/16, pre-trained for 1600 epochs, we achieve +.8%, +.6% and 3.2% improvement for iNaturalist, CIFAR10 and CIFAR100, respectively, while using a model pre-trained for 300 epochs only.

Table 2: Top-1 class. acc. of iNaturalist17, CIFAR10 & CIFAR100 by fine-tuning pre-trained ViT-B/16 (ImageNet-1K).

| Method | Dataset | Pre-training epochs | Fine-tuning Top1 acc % |
|---|---|---|---|
| Supervised (He et al., 2022) | iNaturalist17 | | 68.7 |
| MAE (He et al., 2022) | iNaturalist17 | 1600 | 70.5 |
| ROPIM | iNaturalist17 | 300 | **71.3** |
| MAE (He et al., 2022) | CIFAR10 | 1600 | 97.0 |
| BEiT (Bao et al., 2022) | CIFAR10 | 800 | 96.1 |
| ROPIM | CIFAR10 | 300 | **97.6** |
| MAE (He et al., 2022) | CIFAR100 | 1600 | 82.5 |
| BEiT (Bao et al., 2022) | CIFAR100 | 800 | 80.0 |
| ROPIM | CIFAR100 | 300 | **85.7** |

**Semantic segmentation**. The efficiency of our proposed ROPIM for transfer learning on a segmentation task is presented in Table 3. Following (He et al., 2022; Xie et al., 2022), we run experiments on ADE20K using UperNet (Xiao et al., 2018). Table 3 shows that our pre-training improves results over supervised pre-training, tokenizer-based BEiT, and MAE approaches with less pre-training time.

Table 3: ADE20K semantic segmentation (mIoU) using UperNet (Xiao et al., 2018). (Baselines from MAE (He et al., 2022).)

| Method | Backbone | Pre-training epochs | Fine-tuning mIoU |
|---|---|---|---|
| Supervised (He et al., 2022) | ViT-B/16 | - | 47.4 |
| BEiT (He et al., 2022) | ViT-B/16 | 800 | 47.1 |
| MoCo v3 (He et al., 2022) | ViT-B/16 | 300 | 47.3 |
| MAE (He et al., 2022) | ViT-B/16 | 1600 | 48.1 |
| ROPIM | ViT-B/16 | 300 | **48.5** |

For ADE20K segmentation we use an AdamW optimizer, a weight decay of 0.05, a batch size of 16, and a grid search for learning rate. All models are trained for 160K iterations with an input resolution of 512×512. Following (Bao et al., 2022; Xie et al., 2022) we initialized the segmentation models using model weights after supervised fine-tuning on ImageNet-1K.

### 4.4 VISUALIZATION

Figure 9 displays sample images, their sketched images[2], the predicted images after complement count sketching[3] and the final reconstructed images as "sketched image+predicted complement-sketched image". Note that sketched image is the available data sent to the network. The visible regions represent non-removed information, whereas the corrupt parts are regions where the lossy nature of sketching removed information. In a similar way, the predicted complement space can be seen as information predicted by the network, which was removed from the input

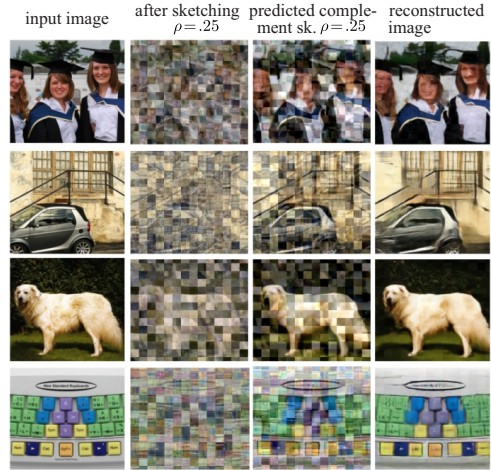

Figure 9: The effect of ROP on images sampled from ImageNet (first column) and patches within. Second column: images after applying Eq. (2) with $\mathbf{W} = \mathbb{I}$, *i.e.*, $\mathbf{P}^\dagger \mathbf{P} \mathbf{X}$ (sketching ratio $\rho = .25$). Third column: images after recovering the complement of Eq. 5, *i.e.*, $(\mathbb{I} - \mathbf{P}^\dagger \mathbf{P}) f_\mathbf{\Theta}(\cdot) \mathbf{W}^*$. Fourth column: reconstructed images (second and third columns added).

---

[1]SSL methods call pre-training and fine-tuning on separate datasets as transfer learning (Grill et al., 2020; Caron et al., 2021a; He et al., 2022).

[2]By "sketched image", we mean ROP was applied along the spatial mode, followed by inverse ROP.

[3]By complement sketching, we mean that we applied the complement of the chosen random projection along the spatial mode, followed by its inverse.

image. As can be seen, combining sketched images with the corresponding recovered complement-sketched images produce the reconstructed images which are visually very close to the original images. The advantage of using ROP is that, as a lossy projection, it is well characterized by the noise variance (the lost information is characterized by the bound on the noise variance). At the same time, the complement of the random subspace enables the recovery of the lost information.

## 4.5 Further Experiments and Ablation Studies

To conduct further experiments and ablation studies, in addition to ViT-S and ViT-B, we train a smaller variant of ViT, ViT-T with ImageNet100. ViT-T is pre-trained for 800 epochs with a batch size of 512. The "Supervised" results for ViT-T are obtained by training the model from scratch with random initialized weights for 800 epochs and a grid search for best performing base learning rate. We apply all data augmentation policies used during our fine-tuning for "Supervised" training.

Table 4: Top-1 classification acc. on ImageNet100. *BEiT tokenizer trained on ImageNet-1K, N/A to other methods.

| Method | Backbone | Fine-tuning Top1 acc % |
|---|---|---|
| DINO (Caron et al., 2021a) | ViT-T/16 | 84.60 |
| MoCo v3 (Chen et al., 2021b) | ViT-T/16 | 82.58 |
| BEiT* (Bao et al., 2022) | ViT-T/16 | 85.32 |
| MAE (He et al., 2022) | ViT-T/16 | 82.58 |
| SimMIM (Xie et al., 2022) | ViT-T/16 | 85.08 |
| Supervised | ViT-T/16 | 82.52 |
| ROPIM, $\ell_1$ loss | ViT-T/16 | **86.43** |
| ROPIM, $\ell_2$ loss | ViT-T/16 | **86.70** |

Table 4 shows performance of our implementation of pre-training and fine-tuning baselines on ImageNet100 dataset with ViT-T as backbone. Note for pre-training on ImageNet100 we had to use the tokenizer trained on ImageNet-1k to report BEiT (Bao et al., 2022) results. Thus, the BeiT result in this case are for reference only. For a fair comparison with other methods using ViT-T, we followed the procedure in (He et al., 2022; Xie et al., 2022) where a grid search for the same set of hyper-parameters is applied for all methods (Bao et al., 2022). For all baselines we run both pre-training and fine-tuning with their default (best performing) setup.

**Sketching ratio $\rho$.** Table 9 of Appendix C shows an ablation study on different values of $\rho$ for different backbones and datasets. We observed that $\rho = \frac{1}{7}$ achieves the highest performance and hence this value was used for all experiments unless otherwise mentioned.

**Increasing number of pre-training and fine-tuning epochs.** Table 1 shows a consistent improvement in performance of ROPIM for ImageNet-1k when increasing the number of pre-training epochs. Top-1 accuracy for ImageNet100 with varying number of pre-training and fine-tuning epochs are shown in Tables 7 and 8 of Appendix C.

**The final gist.** Consider a masking problem with just 2 tokens. For standard binary masking, in total there are $2^2$ unique masking patterns. Assuming 50% masking ratio, that just limits patterns to mere 2. However, if masking has more "continuous" nature (*e.g.*, as ROPIM), one can get for example $\{0\%, 25\%, 50\%, 75\%, 100\%\}$ of original energy preserved per token, which gives $5^2$ unique masking patterns. Under 50% masking ratio (assuming it equals to 50% lost information), that yields $(0\%, 100\%), (25\%, 75\%), (50\%, 50\%), (75\%, 25\%), (100\%, 0\%)$ pattern pairs (5 in total). ROPIM has a similar effect to this toy example, but in addition (i) it provides complementary "unmasking" patterns (ii) with bounded/known variance for the implicitly injected noise as per Properties 1–4.

## 5 Conclusions

We have presented Random Orthogonal Projection Image Modeling (ROPIM), a self-supervised pre-training method for Vision Transformers (ViT). Compared to the popular MIM techniques, ROPIM applies count sketching to project features of patch embeddings along their spatial mode into a random subspace (formed according to the sketch matrix principles) and subsequently retracts them from the subspace into the original feature space. ROPIM incurs minimal computational overheads while providing richer masking-unmasking patterns with a guaranteed bounded variance of the noise, *e.g.*, we can "touch" more tokens spatially-wise than masking to create richer masking patterns. We quantify how much we corrupt these tokens and we have a theoretically guaranteed "unsketching" mechanism in analogy to unmasking. ROPIM does not require customized architecture designs, heavy decoders, or tokenizer networks. We hope ROPIM can inspire further research devoted to improving corruption strategies for self-supervised network pre-training.

## ACKNOWLEDGMENTS

This work was funded by CSIRO's Machine Learning and Artificial Intelligence Future Science Platform (MLAI FSP). MH acknowledges ongoing support from the QUT School of Electrical Engineering and Robotics and the QUT SAIVT (Signal Processing, AI and Vision Technologies) lab.

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

# Pre-training with Random Orthogonal Projection Image Modeling (Appendices)

**Maryam Haghighat**[*,†,††], **Peyman Moghadam**[§,†], **Shaheer Mohamed**[§,†], **Piotr Koniusz**[*,§,‡]
[§]Data61♥CSIRO   [†]Queensland University of Technology   [‡]Australian National University
[†]`name.lastname@qut.edu.au`, [§]`name.lastname@data61.csiro.au`

## ABSTRACT

Below we include remaining experiments and details of our proposed Random Orthogonal Projection Image Modeling (ROPIM). Appendix A presents a comparative analysis of the training cost of ROPIM in contrast to state-of-the-art methods. In Appendix B, we delve into additional experiments related to transfer learning with smaller-scale datasets. Appendix C includes ablation studies on the sketching ratio and the effects of varying pre-training and fine-tuning epochs. Detailed discussion on pre-training and fine-tuning settings are included in Appendix D. Additional works are discussed in Appendix E.

## A   RUNTIMES

Figure 1 compares top-1 accuracy *vs.* total pre-training (PT) time with SOTA methods. For fair comparisons, we used the same resources (8×P100 GPUs) and maximum possible batch size for each method, *i.e.*, GPU memory usage for all methods is 16GB per GPU. Total PT time is time per epoch × number of PT epochs. Table 5 shows more details. As seen, MAE has smaller time per epoch, however, it requires larger number of PT epochs to converge and this results in a less efficient total runtime. We note that MAE (He et al., 2022) removes MASK tokens from the encoder to increase pre-training speed. However, recent methods such as CIM (Fang et al., 2023), Data2vec (Baevski et al., 2022), SimMIM (Xie et al., 2022), and MFM (Xie et al., 2023) retain MASK tokens in the encoder to ensure compatibility with different architectures, including hierarchical ViTs (*e.g.*, Swin) and CNNs. Similarly, our ROPIM offers flexibility to be applied to various architectures. We have additionally incorporated results obtained from the LGP (Jiang et al., 2023) mechanism. LGP (Jiang et al., 2023) combines MIM with contrastive learning in a sequential manner. The initial stage involves training the network with MIM loss functions. Following that, the training process proceeds with contrastive learning, incorporating a learning rate decay strategy. During this phase, the lower layers of the network are assigned a smaller learning rate. LGP employs MAE (He et al., 2022) for MIM and MoCo v3 (Chen et al., 2021) for contrastive learning. We have labeled this by "LGP-MAE" in Figure 1 and Table 5. We followed the same settings, replaced MAE with ROPIM in

---

[*]Corresponding authors.   [††]MH conducted this work during the employment with CSIRO.   PK also in charge of the theory.   The code is available at `https://github.com/csiro-robotics/ROPIM`.

Table 5: Runtimes using the same resources (8×P100 GPUs).

| Method | PT epochs | Memory usage per GPU (GB) | Time per epoch (min) | Total PT time (hour) | Top-1 Acc. % |
|---|---|---|---|---|---|
| MoCo v3 (Chen et al., 2021) | 600 | 16 | 126 | 1260 | 83.2 |
| BeiT (Bao et al., 2022) | 800 | 16 | 63 | 840 | 83.2 |
| MAE (He et al., 2022) | 800 | 16 | 31 | 413 | 83.1 |
| MAE (He et al., 2022) | 1600 | 16 | 31 | 826 | 83.6 |
| MFM (Xie et al., 2023) | 300 | 16 | 47 | 235 | 83.1 |
| CIM (Fang et al., 2023) | 300 | 16 | 120 | 600 | 83.3 |
| CIM (Fang et al., 2023) | 800 | 16 | 120 | 1600 | 83.4 |
| CAN (Mishra et al., 2022) | 800 | 16 | 75 | 1000 | 83.4 |
| CAN (Mishra et al., 2022) | 1600 | 16 | 75 | 2000 | 83.6 |
| ROPIM | 300 | 16 | 47 | 235 | 83.5 |
| ROPIM | 500 | 16 | 47 | 391 | 83.7 |
| ROPIM | 800 | 16 | 47 | 626 | 84.0 |
| LGP-MAE (Jiang et al., 2023) | 1600 (MAE) + 300 (MoCo v3) | 16 | 31 (MAE) + 126 (MoCo v3) | 826 (MAE) + 630 (MoCo v3) | 83.9 |
| LGP-ROPIM | 800 (ROPIM) + 300 (MoCo v3) | 16 | 47 (ROPIM) + 126 (MoCo v3) | 626 (ROPIM) + 630 (MoCo v3) | 84.1 |

the first stage, and continued training of ROPIM with MoCo v3 for 300 epochs. The corresponding results are indicated by "LGP-ROPIM".

## B    TRANSFER LEARNING FOR SMALLER SCALE DATASETS

To further investigate the efficiency of our pre-trained models for transfer learning we run fine-tuning on the pre-trained ViT-T model on ImageNet100 for Flower102, CUB-200, CIFAR10 and CIFAR100 datasets. For fine-tuning CIFAR10/100 on ViT-T we simply up-sample CIFAR image resolutions from $32 \times 32$ to $224 \times 224$.

**Flowers102** (Nilsback & Zisserman, 2008) contains images of 102 fine-grained flower species with 1020 train and 6149 test samples. We pre-process this dataset by center-cropping images and resizing crops to $224 \times 224$. **Caltech-UCSD Birds 200 (CUB-200)** (Wah et al., 2011) is annotated with 200 bird species and contains 5994 training images and 5794 testing images.

Table 6 shows that model pre-trained by ROPIM provides powerful features for transfer learning, outperforming SimMIM (Xie et al., 2022) and MAE (He et al., 2022) baselines in all four datasets.

Table 6: Transfer learning on smaller scale datasets.

| Method | Backbone | Pre-training dataset | Fine-tuning dataset | Top-1 acc % |
|---|---|---|---|---|
| MAE (He et al., 2022) | ViT-T/16 | ImageNet100 | Flowers102 | 50.55 |
| SimMIM (Xie et al., 2022) | ViT-T/16 | ImageNet100 | Flowers102 | 53.00 |
| ROPIM | ViT-T/16 | ImageNet100 | Flowers102 | **54.09** |
| MAE (He et al., 2022) | ViT-T/16 | ImageNet100 | CUB-200 | 53.62 |
| SimMIM (Xie et al., 2022) | ViT-T/16 | ImageNet100 | CUB-200 | 59.01 |
| ROPIM | ViT-T/16 | ImageNet100 | CUB-200 | **61.36** |
| MAE (He et al., 2022) | ViT-T/16 | ImageNet100 | CIFAR10 | 94.96 |
| SimMIM (Xie et al., 2022) | ViT-T/16 | ImageNet100 | CIFAR10 | 95.79 |
| ROPIM | ViT-T/16 | ImageNet100 | CIFAR10 | **96.63** |
| MAE (He et al., 2022) | ViT-T/16 | ImageNet100 | CIFAR100 | 79.55 |
| SimMIM (Xie et al., 2022) | ViT-T/16 | ImageNet100 | CIFAR100 | 80.91 |
| ROPIM | ViT-T/16 | ImageNet100 | CIFAR100 | **81.82** |

## C    FURTHER ABLATIONS STUDIES

### C.1    THE EFFECT OF INCREASING NUMBER OF PRE-TRAINING AND FINE-TUNING EPOCHS

For MIM-based methods in general, training with longer epochs improves their performance (He et al., 2022; Bao et al., 2022). Tables 1 and 7 show the effect of increasing the number of pre-training epochs when the same dataset, ImageNet-1k and ImageNet100, is used during pre-training and fine-tuning. As seen, there is a consistent improvement in classification accuracy with increasing the number of pre-training epochs.

Table 7: The effect of increasing the number of pre-training epochs on top-1 accuracy. ImageNet100 is used for both pre-training and fine-tuning of ViT-T. Models are fine-tuned with 100 epochs.

| Pre-training epochs | Backbone | Dataset | Top-1 acc % SimMIM | Top-1 acc % ROPIM |
|---|---|---|---|---|
| 300 | ViT-T/16 | ImgNet100 | 82.09 | **82.98** |
| 600 | ViT-T/16 | ImgNet100 | 83.32 | **84.20** |
| 800 | ViT-T/16 | ImgNet100 | 85.08 | **86.43** |

Table 8 shows the effect of increasing the number of fine-tuning epochs. ROPIM achieves a classification accuracy of 88.71% on ImageNet100 for 300 fine-tuning epochs which is .71% higher than SimMIM for the same setting.

Table 8: The effect of increasing the number of fine-tuning epochs on top-1 classification accuracy. ImageNet100 is used for both pre-training and fine-tuning.

| Fine-tuning epochs | Backbone | Dataset | Top-1 acc % SimMIM | Top-1 acc % ROPIM |
|---|---|---|---|---|
| 100 | ViT-T/16 | ImgNet100 | 85.08 | **86.43** |
| 200 | ViT-T/16 | ImgNet100 | 87.31 | **87.80** |
| 300 | ViT-T/16 | ImgNet100 | 88.0 | **88.71** |

### C.2 ABLATIONS ON DIFFERENT VALUES OF SKETCHING RATIO $\rho$

As seen in Table 9, $\rho$=.14 achieves better results for different backbones.

Table 9: ROPIM top-1 acc. for pretraining ViT-T, ViT-S and ViT-B with ImageNet100 and ImageNet-1k and using different sketching ratio $\rho$.

| Dataset | Backbone | $\rho$ | Top1 acc % |
|---|---|---|---|
| ImageNet100 | ViT-T/16 | .07 | 86.1 |
| ImageNet100 | ViT-T/16 | .14 | **86.4** |
| ImageNet100 | ViT-T/16 | .25 | 84.4 |
| ImageNet100 | ViT-T/16 | .50 | 85.1 |
| ImageNet-1k | ViT-S/16 | .07 | 81.5 |
| ImageNet-1k | ViT-S/16 | .14 | **81.8** |
| ImageNet-1k | ViT-S/16 | .25 | 81.6 |
| ImageNet-1k | ViT-S/16 | .50 | 81.5 |
| ImageNet-1k | ViT-B/16 | .07 | 83.4 |
| ImageNet-1k | ViT-B/16 | .14 | **83.5** |
| ImageNet-1k | ViT-B/16 | .25 | 83.3 |
| ImageNet-1k | ViT-B/16 | .50 | 83.3 |

## D DETAILS OF PRE-TRAINING AND FINE-TUNING SETUPS

Table 10: Fine-tuning hyper-parameters for BEiT, MAE, SimMIM and ROPIM.

| Config | Value |
|---|---|
| Optimizer | AdamW |
| Weight decay | 0.05 |
| Optimizer momentum | $\beta_1 = 0.9, \beta_2 = 0.999$ |
| Learning rate schedule | cosine decay |
| Warmup epochs | 5 |
| Label smoothing | 0.1 |
| Mixup | 0.8 |
| Cutmix | 1.0 |
| Drop path | 0.1 |
| Rand Augment | 9/0.5 |

Main settings of ROPIM, SimMIM, MAE (He et al., 2022) and BeiT (Bao et al., 2022) are similar for fine-tuning, shown in Table 10. Below we provide additional details.

SETUP OF ROPIM

The models are pre-trained with linear learning rate (lr) scaling rule, lr = base_lr × batch_size/512 is used. Data augmentation strategy includes random *resize cropping* with the scale range of [0.67, 1], an the aspect ratio range of [3/4, 4/3], followed by random *horizontal flipping* and *color normalization*. Pre-training base learning rate for ViT-T is 1e-3, and 1.5e-4 for ViT-S and ViT-B.

During fine-tuning a weight decay of 0.05, $\beta_1 = 0.9$, $\beta_2 = 0.999$, a stochastic depth ratio of 0.1 is employed. We also follow the data augmentation used in (Bao et al., 2022; Xie et al., 2022) and use Mixup, Cutmix, label smoothing, and random erasing. Vit-B and ViT-T models are fine-tuned for 100 epochs unless otherwise mentioned. ViT-S is fine-tuned for 200 epochs. Kindly note that all other baseline methods use the same or longer fine-tuning epochs compared to our work. For fine-tuning, we run a grid search on {5e-3, 1e-2, 2e-2} and report the highest performing one.

In what follows, we provide the pre-training and fine-tuning setup of our ablation studies with ViT-T on ImageNet100. For all baselines, we ran a grid-search on their default learning rate, multiplied by {.1, 1, 2, 4, 10}, and we reported the best performing result. Kindly note that the grid search is a common strategy for selecting hyper-parameters in self-supervised pipelines (Bao et al., 2022).

## SETUP OF SIMMIM

Following the default setting in SimMIM (Xie et al., 2022) for ViT backbone, we used random masking with a patch size of 32×32 and a mask ratio of 0.6, a linear prediction head with a target image size of 224, the $\ell_1$ loss for masked pixel prediction. The models are pre-trained with the AdamW optimizer, a base learning rate of 1e-3, and a multi-step learning rate scheduler with an initial 10 epochs linear warm-up procedure. A linear lr scaling rule, lr = base_lr × batch_size/512 is used. Data augmentation strategy during pre-training includes random *resize cropping* with the scale range of [0.67, 1], an aspect ratio range of [3/4, 4/3], followed by random *horizontal flipping* and *color normalization*.

For fine-tuning, base learning rate of 5e-3 with a layer-wise lr decay of 0.65 is used.

## SETUP OF MAE

For pre-training, MAE (He et al., 2022) uses the cosine learning rate scheduler with a base learning rate of 1.5e-4, the AdamW optimizer with momentum $\beta_1 = 0.9, \beta_2 = 0.95$ and linear lr scaling rule lr = base_lr × batch_size/256. Random-resized *cropping* and *horizontal flipping* are used during pre-training. The features of MAE are extracted from the encoder of the pre-trained network and fine-tuned following the standard supervised ViT training.

For fine-tuning, a base learning rate of 5e-3 and a layer-wise lr decay of 0.75 are used.

## SETUP OF BEIT

Image tokenizer of BEiT (Bao et al., 2022) is adopted from (Ramesh et al., 2021) and the vocabulary size of visual tokens is set as 8192. BEiT uses the AdamW optimizer with a base learning rate of 1.5e-3 for pre-training. We follow their default augmentation policies, *i.e.*, *random-resized cropping*, *horizontal flipping* and *color jittering* for pre-training the network.

For fine-tuning, base lr of 3e-3 and a layer-wise lr decay of 0.65 are used.

## SETUP OF DINO

We pre-train DINO (Caron et al., 2021) with the AdamW optimizer and a base learning rate of 5e-4. We used linear lr scaling rule, lr = base_lr × batch_size/256, with warm up epochs set to 10. For augmentations, *color jittering*, *Gaussian blur*, *solarization* and *multi-cropping* are used following the default setting.

During fine-tuning, we used the pre-trained network with a linear classifier and trained end-to-end for 100 epochs. An SGD optimizer with learning rate of 1e-3 and cosine learning rate decay is used. Random-resized *crop* and random *horizontal flip* augmentations were applied during fine-tuning.

## SETUP OF MOCO V3

We pre-trained MoCo v3 (Chen et al., 2021) with base learning rate of 1.5e-4 and lr scaling rule lr = base_lr × batch_size/256. *Random-resized cropping*, *horizontal flipping*, *color jittering*, *grayscale conversion*, *blurring* and *solarization* were applied following (Chen et al., 2021).

We fine-tuned the pre-trained model end-to-end for 100 epochs following DEiT (Touvron et al., 2021) setup. We used the official implementation of MoCo v3 to convert the pre-trained model to DEiT supporting format to perform fine-tuning. Here, we used a learning rate of 5e-4, an AdamW optimizer with cosine learning rate decay and 5 warm-up epochs. Linear lr scaling rule, $lr = base\_lr \times batch\_size/512$ is used.

We applied the default augmentations in DEiT which include *color jitter*, *rand_augment = 9/0.5*, *mixup_prob = 0.8*, *cutmix_prob = 1.0* and *erasing_prob = 0.25*.

## E    MORE RELATED WORKS

Self-supervised learning is an active research area. Researchers from different fields have been creative in using MIM techniques for various applications such as video (Wang et al., 2023b), point clouds (Tian et al., 2023) and hyper-spectral images (Mohamed et al., 2024). However, they are not directly related to our work as they research other modality types. Combining multiple pre-training strategies and data from various modalities/sources can also greatly boost the training of large-scale models (Su et al., 2023). Another very recent pipeline, Correlational Image Modeling (CorIM) (Li et al., 2023a), proposes a self-supervised pre-training task leveraging a cropping strategy, a bootstrap encoder, and a correlation decoder. However, the performance of CorIM (83.1% top 1 accuracy for ViT-B) appeared short of the performance of pure ROPIM 83.5% with 300 pre-training epochs. Along with masked image modeling, other strategies utilize multiple contrastive heads (Wang et al., 2023a). Different with masked modeling, noteworthy is also abundance of other self-supervised strategies applied in self-supervised 2D/3D feature matching for re-localization (Ramezani et al., 2023), image deblurring (Zhang et al., 2023a), image-to-image translation (Shiri et al., 2019), segmentation self-distillation (Kang et al., 2023), GAN consistency regularization (Ni & Koniusz, 2023), keypoint contrastive learning (Lu & Koniusz, 2024), categorical data learning (Li et al., 2023b), traffic predictive coding (Prabowo et al., 2023a;b), multi-language output self-supervision (Tas & Koniusz, 2021) and graph contrastive collaborative learning (Zhang et al., 2023b).

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
