# OpenReview forum: "Pre-training with Random Orthogonal Projection Image Modeling"
_ICLR.cc/2024/Conference — ICLR 2024 spotlight_

### Official Review · Reviewer_zDzQ · 2023-10-28

**Soundness:** 3 good
**Presentation:** 3 good
**Contribution:** 3 good
**Rating:** 6
**Confidence:** 3

**Summary:**

This paper focuses on the task of MIM, by designing a new corruption process via the random orthogonal projection. Such a ROP strategy results in a more efficient and effective pre-training method. Another advantage of ROP is the guaranteed bound on the noise variance during the corruption process.

**Strengths:**

1. This work is based on the sound theory of random orthogonal projection.
2. ROPIM is able to achieve more superior performance in a shorter pre-training time.
3. The decoder only contains one linear layer, being more slight comparing with MIM.
4. The experiments verify its effectiveness on several downstream tasks, including classification and segmentation.

**Weaknesses:**

1. It is a little difficult to understand why does the proposed method is more superior than MIM. According to my understanding, the ROP strategy randomly discards some local (not global) patterns during corruption as shown in Fig. 9. This is very similar to MIM. So, I can't intuitively catch what results in the superiority of ROP in the field of MIM. It would be better to have a discussion in the paper.

**Questions:**

1. Proposition 1 gives a complicated way to generate a projection matrix $P$ that contains only three elements, namely {-1, 0, 1}. This procedure relies on two auxiliary variables of $h$ and $s$. Is it possible to directly sample from {-1, 0, 1} for each entry of $P$?

2. Following question 1, the projection matrix $P$ is composed of {-1, 0, 1}. Generally speaking, "0" denotes discarding some embed patch in projection. It is hard to understand the function of "-1".

3. In proposition 1, I guess: $h \in I^d_{K^{'}}$ should be $h \in I^K_{K^{'}}$?

4. In proposition 2, the notation for $\phi^{'}$ is missing.

---

> ### Author Response · Authors · 2023-11-20
> **Rebuttal**
>
> # Response to Rev. 3 (Reviewer zDzQ)
>
> *Firstly, **we thank the reviewer** for the constructive review and valuable questions.*
>
> ## 1. Little difficult to understand why the proposed method is better than MIM. The ROP strategy randomly discards some local (not global) patterns during corruption as shown in Fig. 9. This is very similar to MIM.
>
> Thank you. The simplified answer is that we achieve better results due to different profile/type of noise we implicitly inject via sketching  compared to binary masking:
> * **masking introduces binary patterns** (keep token or completely mask it) resulting in a limited number of masking patterns
> * **Fig. 8 and 9 show that** rather than just removing token, **our ROPIM has this more continuous effect compared to binary masking**, which creates more patterns of randomness (Figure 4 compares the effect of binary masking versus  our strategy).
>
> For simplicity, **imagine a masking problem with just 2 tokens**:
> * **for standard binary masking, in total there are $2^2$ unique masking patterns**, and assuming 50\% masking ratio, that just **limits patterns to mere 2**.
> * in contrast, if masking has more \'continuous\' nature, **say we can get $\\{0\\%, 25\\%, 50\\%, 75\\%, 100\\%\\}$ of original energy preserved per token, that gives $5^2$ unique masking patterns**, and under 50\% masking ratio (assuming it equals to 50\% lost information), that gives us (0,100),(25,75), (50,50), (75,25), (100,0) patterns (5 in total). While this is a simplification, ROPIM yields a similar effect but (i) in addition we get complementary \`unmasking\' pattern and known variance bound of the implicitly injected noise as per Prop. 1-5.
>
>
> **Importantly, in Figure 7a we show the actual real-data comparison between our ROPIM strategy and binary masking**. Figure 7a illustrates numerically what we argue above, i.e., for the same \`masking ratio\':
> * **binary masking makes many tokens at the input to be completely unmodified** (see the first histogram bean in Fig. 7a)
> * in contrast, our **ROPIM leaves fewer tokens \`untouched\' by some level of noise - creating richer noise patterns than binary masking**
> * consequently, Figure 7d also shows that our \`unmasking\' step works much harder
>
> Thus, **ROPIM leads to greater variety of input patterns and \`unmasking\' patterns**, which exposes network to greater diversity of patterns to help learn the implicit manifold of the data (think e.g. DAE).
> \
> \
> Note that suitable data augmentation strategies during pre-training are an open question:
> > [I]. Masked auto-encoders are scalable vision learner, He *et al.*, CVPR 2022
>
> > [II]. Mixed auto-encoder for self-supervised visual representation learning, Chen *et al.*, CVPR 2023
>
> > [III]. Corrupted image modeling for self-supervised visual pre-training. Fang *et al.*, CVPR 2023.
>
> E.g., color jittering degrades the transfer learning performance of MAE in [III], implying that MIM exhibits a  preference for specific kinds of data augmentation (not all are good).
>
>
> ## 2. Proposition 1 gives a complicated way to generate a projection matrix with elements: -1, 0, 1. This procedure relies on h and s. Is it possible to directly sample from -1, 0, 1 for each entry of P ?
>
> Thank you. We appreciate the process may look complicated but this is the standard definition of count sketching. Both variables are necessary because hashing function h (producing ${\bf h}$ ) controls unique locations of non-zero entries per row of ${\bf P}$ and thus promotes each row vector of ${\bf P}$ to be orthogonal to other row vectors. **This is essential in order to ensure we deal with subspaces (and that Prop. 2-5 hold).**
>
> ## 3. The projection matrix P is composed of -1, 0, 1. ”0” denotes discarding some embed patch in projection. It is hard to understand the function of ”-1”.
>
> Attributing roles to -1 and 1 at the coefficient level is hard as ${\bf P}$ is a projection matrix for matrix-vector multiplication, not element-wise tool.
>
> The meaning of elements 1 and -1 is that they are an output of a hash function $s$ (${\bf s}$ is specific result from hashing function s).
>
> For detailed explanation of count-sketching, the following article is a sensible introduction to this advanced topic:
> https://en.wikipedia.org/wiki/Count_sketch
>
>  Kindly notice **what matters to ROPIM is the resulting properties in Propositions 2-5** of our work:
> * controllable/bounded variance of approximation
> * easy \`unsketching\' operation (project to subspace and retract back, notice we use ${\bf P^\dagger P \phi}$ on input)
> * easy complement operation for unmasking (output of decoder) that tries recover exactly what was removed on input
> * fast generation and computation (generating ${\bf P}$ is fast, multiplying with ${\bf P}$ too).
>
> ## 4. In prop. 1, $h \in I_{K^{'}}^{d}$ should be $I_{K^{'}}^{K}$?
>
> Thank you. We have now fixed it.
>
> ## 5. Notation $\phi'$ missing.
>
> Our apology: should be  $\phi_y$ not $\phi'_y$. $\phi_x$ and $\phi_y$ reads as \`take any two vectors\'.

---

> > ### Comment · Reviewer_zDzQ · 2023-11-21
> > **Response authors**
> >
> > Thanks for your response. This response addressed all of my concerns. I remain my postive rating.
> >
> > I suggest to add the discussion in Q1 to the final version.

---

> > > ### Author Response · Authors · 2023-11-21
> > > **thank you**
> > >
> > > Than you. Absolutely, we will add such a discussion.
> > > If there is anything more we can improve or answer meantime, kindly let us know.
> > >
> > > Best regards,
> > > \
> > > Authors

---

### Official Review · Reviewer_xoq6 · 2023-10-31

**Soundness:** 3 good
**Presentation:** 3 good
**Contribution:** 3 good
**Rating:** 6
**Confidence:** 4

**Summary:**

This paper considers the self-supervised learning problem, which has attracted much attention recently. While the masked image modeling, e.g. masked autoencoder, has recently shown very promising performance in self-supervised learning for visual pre-training, the authors aim to improve it by proposing a new image modeling. Specifically, unlike the masked image modeling applying random crops to the input and learning to recover the masked inputs with an encoder-decoder network, this paper considers a random orthogonal projection modeling which uses a random subspace for the projection and then learns to recover the complement of that subspace. Provided experiments show that the proposed random orthogonal projection can yield better performance than the crop-based masking.

**Strengths:**

Though simple and straightforward, to my knowledge the proposed random orthogonal projection modeling for self-supervised learning is novel. Provided experimental results have demonstrated better performance of the proposed random orthogonal projection method in comparison with the masked image modelling using crop-based masking.

**Weaknesses:**

The proposed method is somewhat heuristic.

**Questions:**

Some of the Propositions in Section 3.1 are rather straightforward and would be better not be expressed as Proposition.

$\ell_1$ loss is used in the reconstruction loss, would the MSE loss yield worse performance?

---

> ### Author Response · Authors · 2023-11-20
> **Rebuttal**
>
> # Response to Rev. 2 (Reviewer xoq6)
>
> *Firstly, **we thank the reviewer** for the constructive review and valuable questions.*
>
> ## 1. The proposed method is somewhat heuristic.
>
> Thank you. The very standard Masked Auto Encoder (MAE) and related works perform basic binary masking which we agree may be viewed as heuristic. However, MAE can be inspired by the mathematics of Denoising Auto-encoders:
> > What regularized auto-encoders learn from the data-generating distribution., Alain and Bengio, JMLR 2014
>
> The above study explains how the variance of noise injected into DAE is directly connected with the gradient regularization and manifold learning property, e.g., consider Reconstruction Contractive Auto-encoder (RCAE) (their Eq. 6 and Section 3.1)
> \
> \
> From that point of view, transformer-based MAE and related works may be harder to analyze but **if we can quantify the variance of the noise we inject, we can lean on the theory of DAE and RCAE to understand how the noise variance governs their data manifold learning property.**
> \
> \
> Thus, our **Proposition 2 we quantify the upper bound on variance implicitly introduced by our projection in ROPIM**. From that point of view, **our ROPIM enjoys better foundations than existing masked auto-encoders** as we are able to quantify the injected variance while previous works do not seem to notice the importance of noise variance, and they do not provide any guarantee or indications in that sense at all.
> \
> \
> We will ensure to explain that theoretical connection better in our final draft.
>
> ## 2. Some of the Propositions in Section 3.1 are rather straightforward and would be better not be expressed as Proposition.
>
> Thank you. We are more than happy to simply rephrase propositions 2-5 as a list of **properties** of ROPIM with sketch-based projections. We are more than keen to read further advice from the reviewer.
> \
> \
> While simple, we include Property 2 as it explains the variance of implicitly injected noise (important due to the conenction to DAE), Propositions 3 and 4 explain how to use such projections in practical terms (e.g., unsketchign is needed to keep tokens in one common space), Proposition 5 is important as it gives \`the complement of sketching\' operation required to guide the output of our approach, and ensures it \`unmasks\' exactly the information that was removed at the input.
>
>
>
>
> ## 3. The $\ell_1$ loss is used in the reconstruction loss, would the MSE loss yield worse performance?
>
> Thank you for spotting this.  We have now conducted experiments with MSE for ViT-T and ImageNet100, and the MSE loss  performs slightly better than the $\ell_1$ loss. We will include such additional ablations.
>
>
> |Loss | Top 1 acc. (with 300 PT epochs) | Top 1 acc. (with 800 PT epochs)  |
> |-|-|-|
> |ROPIM with $\ell_1$ | 82.98 | 86.43 |
> |ROPIM with MSE | **83.60** | **86.70** |

---

### Official Review · Reviewer_5QU4 · 2023-10-31

**Soundness:** 3 good
**Presentation:** 3 good
**Contribution:** 3 good
**Rating:** 8
**Confidence:** 4

**Summary:**

The work proposes to use random projections in the place of masked images for pre-training the ViTs. The work is shown to lead to better performance for classification tasks compared to the MIM methods. The work seems novel enough where they replace binary mask with floating point mask.

**Strengths:**

+ The use of linear algebraic projection technique in both the method and the loss function for reconstruction.

+ The results are achieved better with a considerably smaller number of epochs.

+ The work is well-motivated from the basics and seems reproducible.

+ The transfer learning results are added value to the work as such.

+ The work should be useful as a pre-trainer for several ViT based applications.

**Weaknesses:**

- I'm not sure if all the recent works on MIM have been compared with. Authors are requested to comment on this.

**Questions:**

Are there any recent works which have been exempted from comparison?

Apart from classification and semantic segmentation, do the authors have results on any other applications?

---

> ### Author Response · Authors · 2023-11-20
> **Rebuttal**
>
> # Response to Rev. 1 (Reviewer 5QU4)
>
> *Firstly, **we would like to sincerely thank the reviewer** for the constructive review and valuable questions.* Below are point-by-point responses.
>
> ## 1. Are there any recent works which have been exempted from comparison? I am not sure if all the recent works on MIM have been compared with.
>
> * Self-supervised learning is an active research area. We have tried our best to cover all related works. Researchers from different fields have been creative in using MIM techniques for various applications such as video:
> >[I]. Videomae v2: Scaling video masked autoencoders with dual masking, Wang *et al.*, CVPR 2023
>
>   and point clouds:
>   >[II]. Geomae: Masked geometric target prediction for self- supervised point cloud pre-training, TIan *et al.*, CVPR 2023
>
>   However, they are not directly related to our work as they research other modality inputs.
>
> * Another stream of research studies merging various pre-training strategies, i.e., combining contrastive learning with MIM, using teacher-student networks or utilizing networks pre-trained  on large multi-modal datasets:
>   >[III]. Towards all-in-one pre-training via maximizing multi-modal mutual information, Su *et al.*, CVPR 2023
>
>   Such ideas can be potentially integrated with ROPIM but they are orthogonal to our work, and training each such a pipeline takes several days if not weeks. Nonetheless, we have equipped the most promising in our modest view such a framework, called **LGP** [IV] (in Fig. 1 we called it **GPL** due to typo), with ROPIM and showed substantial improvement in our Fig. 1.
>   > [IV]. Bridging contrastive learning and masked image modeling for label-efficient representations
>
>   Another very recent pipeline, called Correlational Image Modeling (CorIM) [V], proposes a self-supervised pre-training task leveraging a cropping strategy, a bootstrap encoder, and a correlation decoder. However, performance of CorIM (83.1\% top 1 acc. for ViT-B) is short of the performance of pure ROPIM's **83.5\%** with 300 pre-training epochs. Thus, as the most promising new strategy, we investigated GPL with our ROPIM (GPL+ROPIM in Fig. 1 of main submission).
>   >[V].  Correlational image modeling for self-supervised visual pre-training, Li *et al.*, CVPR 2023
>
>   We also reported recent CIM models [VII] which also perform worse than ours despite using a mix of contrastive and masking strategies (e.g., 81.6 vs. ours 82 in Table 1).
>
>
> ## 2. Apart from classification and semantic segmentation, do the authors have results on any other applications?
>
> Following the very recent works [IV, V, VI, VII], we have reported results on classification and segmentation downstream tasks as each downstream task is time-consuming (e.g., 4 days for segmentation).
> > [VI]. Masked frequency modeling for self-supervised visual pre-training, Xie *et al.*, ICLR 2023
>
>   > [VII]. Corrupted image modeling for self- supervised visual pre-training, Fang *et al.*, ICLR 2023
>
> Nonetheless, we have also initiated less often used object detection on COCO for ViT-B. After 5 days of running the code we have  completed 50/100 epochs with 50.01\% bbox/AP while MAE completed 50/100 epochs at 49.19\%. We anticipate to reach 100 epochs within another 5 days on our 8 GPU server. We will update our final manuscript accordingly.

---

> > ### Comment · Reviewer_5QU4 · 2023-11-21
> >
> > I am satisfied with the response by the authors. Retain my positive rating.

---

> > > ### Author Response · Authors · 2023-11-21
> > > **thank you**
> > >
> > > Thank you,
> > >
> > > If there is anything meantime we can add, change or improve, kindly do let us know.
> > >
> > > Best regards,
> > > \
> > > Authors

---

### Author Response · Authors · 2023-11-14
**rebuttal coming soon**

Esteemed Reviewers,
\
\
We want to thank for your constructive comments which help us refine our work.

We just wanted to let you know that **we are working on the rebuttal right now** to answer all your questions. We hope to post detailed answers around weekend.
\
\
Kind regards,
\
Authors

---

### Meta-Review · Area_Chair_5K1a · 2023-12-01

**Metareview:**

The paper propose an Image Modeling framework based on random orthogonal projection instead of binary masking as in MIM. The work is well-motivated from the basics and seems reproducible. Provided experimental results have demonstrated the superiority of the proposed random orthogonal projection method.

The reviewer has mentioned that It’s a little difficult to understand why does the proposed method is more superior than MIM. Perhaps adding a little discussion in the paper would be better.

**Justification For Why Not Higher Score:**

The acceptance of the paper has been voted on.

**Justification For Why Not Lower Score:**

The idea is novel and  based on the sound theory.

---

### Decision · Program_Chairs · 2024-01-16

Accept (spotlight)